# Gallstone Disease, Obesity and the Firmicutes/Bacteroidetes Ratio as a Possible Biomarker of Gut Dysbiosis

**DOI:** 10.3390/jpm11010013

**Published:** 2020-12-25

**Authors:** Irina N. Grigor’eva

**Affiliations:** Laboratory of Gastroenterology, Research Institute of Internal and Preventive Medicine-Branch of The Federal Research Center Institute of Cytology and Genetics of Siberian Branch of Russian Academy of Sciences, Novosibirsk 630089, Russia; niitpm.office@gmail.com; Tel.: +7-9137520702

**Keywords:** Firmicutes, Bacteroidetes, gut microbiota, bile microbiota, gallstone patients

## Abstract

Obesity is a major risk factor for developing gallstone disease (GSD). Previous studies have shown that obesity is associated with an elevated *Firmicutes/Bacteroidetes* ratio in the gut microbiota. These findings suggest that the development of GSD may be related to gut dysbiosis. This review presents and summarizes the recent findings of studies on the gut microbiota in patients with GSD. Most of the studies on the gut microbiota in patients with GSD have shown a significant increase in the phyla *Firmicutes* (Lactobacillaceae family, genera *Clostridium*, *Ruminococcus*, *Veillonella*, *Blautia*, *Dorea*, *Anaerostipes*, and *Oscillospira*), *Actinobacteria* (*Bifidobacterium* genus), *Proteobacteria*, *Bacteroidetes* (genera *Bacteroides*, *Prevotella*, and *Fusobacterium*) and a significant decrease in the phyla *Bacteroidetes* (family *Muribaculaceae*, and genera *Bacteroides*, *Prevotella*, *Alistipes*, *Paludibacter*, *Barnesiella*), *Firmicutes* (genera *Faecalibacterium*, *Eubacterium*, *Lachnospira*, and *Roseburia*), *Actinobacteria* (*Bifidobacterium* genus), and *Proteobacteria* (*Desulfovibrio* genus). The influence of GSD on microbial diversity is not clear. Some studies report that GSD reduces microbial diversity in the bile, whereas others suggest the increase in microbial diversity in the bile of patients with GSD. The phyla *Proteobacteria* (especially family *Enterobacteriaceae*) and *Firmicutes* (*Enterococcus* genus) are most commonly detected in the bile of patients with GSD. On the other hand, the composition of bile microbiota in patients with GSD shows considerable inter-individual variability. The impact of GSD on the *Firmicutes*/*Bacteroidetes* ratio is unclear and reports are contradictory. For this reason, it should be stated that the results of reviewed studies do not allow for drawing unequivocal conclusions regarding the relationship between GSD and the *Firmicutes*/*Bacteroidetes* ratio in the microbiota.

## 1. Introduction

Obesity is defined as excessive fat accumulation that may impair health; obesity is a result of an imbalance between energy intake and expenditure [1,2]. Today obesity has become pandemic; about 1.9 billion people on the planet are overweight: overall, about 13% of the world’s adult populations (11% of men and 15% of women) were obese in 2016 [3]. The World Health Organization (WHO) estimated that nearly 2.8 million deaths annually are a consequence of overweight and obesity-associated conditions [3], such as atherosclerosis, diabetes, gallstone disease (GSD), etc. [4,5,6].

GSD is a common benign gastrointestinal disease affecting 10–15% of adults around the world that greatly contributes to health care costs [7,8,9,10]. Risk factors of the GSD are age, female sex, obesity, insulin resistance, physical inactivity, genetic background, dietary factors (high carbohydrate, high-calorie intake), dyslipoproteinaemia, certain diseases (such as diabetes mellitus, nonalcoholic fatty liver disease (NAFLD), hypertension, and cardiovascular disease) and medications (hormone replacement therapy, fibrates, etc.), social and economic issues, fertility, and intestinal factors (with increased absorption of cholesterol, slow intestinal motility, and dysbiosis) [7,8,9,10]. Obesity is a major risk factor for developing GSD [9,10,11,12] because it is accompanied by increased synthesis and excretion of cholesterol into bile [13], wherein the amount of cholesterol produced is directly proportional to being overweight [11].Obesity is regarded as an inflammatory condition [14]. Inflammation may be the potential link between insulin resistance and gallstones [15]. Insulin resistance is considered a risk factor for GSD, as it may lead to excess biliary cholesterol production and saturation [16,17] and alone may be responsible for gallbladder dysmotility [18]. However, the absence of a relationship between body mass index (BMI) and GSD had been reported in several epidemiologic studies [7,8,19]. The possible pathogenesis for the close association between obesity and GSD iscomplex and not fully understood.

A significant relationship exists among food intake, energy balance and gut peptides that are secreted from gastrointestinal enteroendocrine cells, such as ghrelin, leptin, glucagon-like peptide-1, cholecystokinin (CCK), peptide tyrosine tyrosine (PYY), and serotonin [20]. Let’s focus on two of them. Ghrelin, an orexigenic peptidyl hormone secreted from the stomach, was discovered in 1999 and is associated with feeding and energy balance [21]. Ghrelin increases appetite and energy expenditure and promotes the use of carbohydrates as a source of fuel at the same time as sparing fat [22]. The development of resistance to leptin andghrelin, hormones that are crucial for the neuroendocrine control of energy homeostasis, is a hallmark ofobesity [23]. The impact of acyl-ghrelin on glucose metabolism and lipid homeostasis may allow for novel preventative or early intervention therapeutic strategies to treat obesity-related type 2 diabetes and associated metabolic dysfunction [24]. There were no differences for total bile acids, insulin, ghrelin, and glucose-dependent insulinotropic polypeptide between patients with GSD and the control group without gallstones [25]. Mendez-Sanchez et al. (2006) found an inverse correlation of serum ghrelin levels and theprevalence of GSD in alogistic regression analysis (OR = 0.27, 95% CI 0.09–0.82, *p* = 0.02) [26]. Authors suggest that serum ghrelin concentrations are associated with a protective effect of GSD and this is related to a motilin-like effect of ghrelin on the gallbladder motility. However, themedian of serum ghrelin values did not show a difference between the patients and controls (660 vs. 682 ng/L) [26].

Leptin is associated with obesity: although it should reduce food intake and body weight, in obese patientsthe serum leptin levels are higher than in the lean individuals and do not manage reducing their food intake [27]. Insulin and leptin play an important role in the development of prediabetes and NAFLD, which is a risk factor for GSD. There could be the following pathogenic links: obesity promotes insulin resistance; high levels of insulin increase leptin levels; leptin cannot lead to decreased insulin levels and decreased appetite because of leptin resistance in the nervous system and the adipose tissue; and high levels of leptin promote hepatic steatosis which in turn increases insulin resistance [27]. Positive correlations between serum leptin and BMI, CCK, total cholesterol, and insulin were found in the gallstone group [28].

Gut microbiota can regulate levels of these gut peptides and thus regulate intestinal metabolism via the microbiota-gut-brain axis [20]. Serum ghrelin levels were negatively correlated with *Bifidobacterium, Lactobacillus* and *B. coccoides–Eubacterium rectale*, and positively correlated with *Bacteroides* and *Prevotella* [29]. Leptin was negatively correlated with *Clostridium, Bacteroides* and *Prevotella*, and positively correlated with *Bifidobacterium* and *Lactobacillus* [29].The results of the studies on the relationship between GSD, obesity, and incretin hormones remain controversial.

GSD and obesity have similar prevalence [10]. Most of the above risk factors are common to GSD and obesity. Despite the increasing number of scientific publications on the gut microbiota in obesity, there is a lack of studies that assess the gut microbiota in GSD. Research on this topic is limited and mainly focused on the study of certain genera and species of microorganisms, but not the *Firmicutes/Bacteroidetes* ratio. Many studies have shown that, in humans, obesity is associated with an increased *Firmicutes/Bacteroidetes* ratio in comparison with lean or “healthy obese” individuals [1,2,30,31,32,33,34,35,36,37,38,39,40,41,42]. This review presents and summarizes the recent findings of studies on the gut microbiota in patients with GSD regarding the *Firmicutes/Bacteroidetes* ratio, as a possible biomarker of obesity, given that obesity is a key risk factor for GSD.

### The Gut Microbiome and Its Functions

Bacteria emerged 3.8 billion ago [43]. There are about 10 trillion human cells in the human body and about 100 trillion cells outside and inside our bodies being of microbial origin [44,45]. The gut microbiome is a dynamic assembly of microorganisms and the resultant products of their collective genetic and metabolic materials, containing from 2 to 20 million microbial genes by the human microbiome’s predominantly in the gut [44]. The gut microbiome plays an array of biological functions ranging from controlling gut–immune system axis, providing several key metabolites and maintaining an optimal digestive system due to the presence of genes, which encode digestive enzymes that are not present in human cells but are associated with the metabolism and fermentation of many food compounds necessary for the host’s nutrition [46]. A greater richness and diversity of bacterial species in the human intestine may be an indicator of health [45,47,48].

## 2. The Firmicutes/Bacteroidetes Ratio

### 2.1. Short Characteristics of the Firmicutes and the Bacteroidetes

As the dominant gut microbiota in healthy adult humans [4], intestinal bacteria include members of both the *Firmicutes* (range of quantitative data–20.5% up to 80% [49,50,51,52,53]) and the *Bacteroidetes* (from 13.85% up to 75.3% [20,49,51,52]). Major taxa of the *Firmicutes* to be included of more than 200 genera [53,54]. *Proteobacteria, Fusobacteria, Actinobacteria, Cyanobacteria*, and *Verrucomicrobia* phyla also are present as minor players [54].

Some *Bacteroides* spp. and *Prevotella* spp. have a variety of glycans and glycosidases that can utilize polysaccharides [53,55]. Other important functions of *Bacteroides* spp. include deconjugation of bile acids [56]. The gut microbiota, especially *Bacteroides intestinalis*, and to a certain extent *Bacteroides fragilis* and *E. coli*, also has the capacity to deconjugate and dehydrate the primary bile acids and convert them into the secondary bile acids in the human colon [57]. Bacteria belonging to the phylum *Bacteroidetes* have high functional redundancy, whereas the phylum *Firmicutes* was comprised of a large number of more functionally diverse core bacteria [53,54,58]. Commensal *Clostridial* clusters XIVa and IV plays an important role in the host and gut homeostasis from the metabolic point of view through the production of short-chain fatty acids, normalizes intestinal permeability, involved the brain–gut axis regulation, in the immune system development, etc. [59]. Many *Firmicutes’* abilities are related to the host’s body weight: obesity-associated gut microbiota is enriched in *Clostridium leptum* [54], Roseburia intestinalis, Eubacterium ventriosum, *Eubacterium hallii* [60], *Lactobacillus reuteri* [42], *Blautia hydrogenotorophica*, *Coprococcus catus*, *Ruminococcus bromii*, *Ruminococcus obeum* [50]. However, other *Firmicutes* are abundant in non-obese individuals: *Clostridium cellulosi,* associated with the degradation of plant material [60,61], *Clostridium orbiscindens* (currently known as *Flavonifractor plautii*), capable of utilizing flavonoids [52], *Clostridium bolteae*, *Blautia wexlerae* [58], *Clostridium difficile*, the *Staphylococcus* genus [40], *Oscillospira guillermondii* [60], *Faecalibacterium (prausnitzii), Lactobacillus plantarum,* and *paracasei* [42]. Also, two *Bacteroides* species (*B. faecichinchillae* and *B. thetaiotaomicron*) [58] and *Akkermansia muciniphila,* and *Methanobrevibacter smithii* [42] were significantly more abundant in stool samples from non-obese compared with obese subjects. Such differences in the “behaviour” of bacteria cannot be explained only by their metabolic properties, because of the exact functions of bacteria are still unclear.

### 2.2. The Story of “Discovery” of the Firmicutes/Bacteroidetes Ratio

Increased efficiency of energy harvest, due to alterations in the gut microbiota has been implicated in obesity in mice [31,32,62] and humans [38].Alterations affecting the dominant intestinal phyla the *Firmicutes* and the *Bacteroidetes* were first described by Ley et al. (2005) in obese animals [1]. In the analysis of the cecal microbiota (by the 16S rRNA gene sequences) of genetically obese ob/ob mice, lean ob/+ and wild-type +/+ siblings, ob/ob animals have a 50% reduction in the abundance of Bacteroidetes and a proportional increase in Firmicutes compared with lean mice [1]. The authors also pointed out that an increase ofthe *Firmicutes*/*Bacteroidetes* ratio may help promote adiposity in *ob*/*ob* mice. The *Firmicutes/Bacteroidetes* ratio is also under debate as a possible biomarker of obesity and related dysfunctions [53,62,63,64,65,66]. A low *Firmicutes/Bacteroidetes* ratio was found to be associated with lean phenotypes, younger age, cardiovascular health, and a balanced immune system and is generally considered beneficial for health [67,68,69].

### 2.3. The Firmicutes/Bacteroidetes Ratio in Obesity: Pro

Ley et al. (2006) have shown that the microbiota in obese subjects shows an elevated proportion of the *Firmicutes* and a reduced population of the *Bacteroides.* Conversely, the relative proportion of the *Bacteroidetes* decreased in humans on a weight-loss program [30]. 16S rRNA gene sequencing revealed a lower proportion of *Bacteroidetes*, more *Actinobacteria* in obese versus lean individuals, but no significant difference in *Firmicutes* in 31 monozygotic twin pairs and 23 dizygotic twin pairs [33]. Armougom et al. [34] confirmed a reduction in the *Bacteroidetes* community in 20 obese patients compared with 20 normal-weight individuals (*p* < 0.01). Zuo et al. (2011) reported that obese people had fewer cultivable *Bacteroides* than their normal-weight counterparts [37]. In the gut in obese adolescents, the total microbiota was more abundant on the phylum *Firmicutes* (94.6%) as compared with *Bacteroidetes* (3.2%) [39]. In the systematic review (PubMed: 2005–2017) adecrease in the *Bacteroidetes* phylum and *Bacteroides/Prevotella* groups was related to high BMI and the *Firmicutes* phylum was positively correlated with weight gain in children between 0 and 13 years of age [40]. In an adult Ukrainian population, the *Firmicutes/Bacteroidetes* ratio was significantly associated with BMI (OR = 1.23, 95% CI 1.09–1.38) and this association continued to be significant after adjusting for confounders such as age, sex, smoking and physical activity (OR = 1.33, 95% CI 1.11–1.60) [41]. The recent systematic review confirmed that individuals with obesity have a greater the *Firmicutes/Bacteroidetes* ratio, more *Firmicutes, Fusobacteria, Proteobacteria, Mollicutes,* and less *Bacteroidetes* [42].

### 2.4. The Firmicutes/Bacteroidetes Ratio in Obesity: Contra

However, other human trials not only failed to confirm a high proportion of *Firmicutes* in obese patients [63,70,71,72,73,74,75,76,77,78] and, but reported even the opposite: about higher amounts of *Bacteroidetes*, and decreased amounts of *Clostridium* cluster XIVa in obese subjects as compared with lean donors [71]. Proportions of the genus *Bacteroides* were greater in overweight volunteers than lean and obese volunteers and the *Firmicutes/Bacteroidetes* ratio changed in favour of the *Bacteroidetes* in overweight and obese subjects [72]. Duncan et al. (2008) found that weight loss did not change the relative proportions of the *Bacteroides* spp, or the percentage of the *Firmicutes* present, in the human gut [73]. In another study, no significant differences in the *Firmicutes/Bacteroidetes* ratios were found between obese and normal-weight adults [74] or obese and normal-weight children [75]. Two meta-analyses have shown that the content of the *Firmicutes* and the *Bacteroidetes* and their ratio is not a consistent feature distinguishing lean from obese human microbiota generally [76,77].

Many authors have concluded that there is no simple taxonomic signature of obesity in the microbiota of the human gut and that significant technical and clinical differences exist between published studies [63] and that the phylum level difference of the gut microbiota between obese and lean individuals might not be universally true [78]. Likely explanations for these controversies are discussed below.

## 3. Role of the Microbiota in the Pathogenesis of Gallstone Disease

The pathogenesis of cholesterol GSD is multifactor, it is determined by five primary defects: genetic background and LITH genes, hepatic hypersecretion of biliary cholesterol, rapid precipitation of solid cholesterol crystals in bile, gallbladder dysmotility, and intestinal factors (with increased absorption of cholesterol, slow intestinal motility, and dysbiosis) [10].

In recent years, attention has been focused on the potential impact of the gut microbiota on the pathogenesis of pigment and cholesterol gallstones. It is proved that intestinal dysbiosis makes a significant contribution to the development of not only the GSD itself [5,6,79,80,81,82], but also to the development of numerous disorders that are risk factors for GSD, including obesity [31,32,33,34,35,36,37,38,39,40,41,42], type 2 diabetes [83], hypercholesterolemia [20,52], diet [84], NAFLD [85,86,87,88], cardiovascular diseases [68,89], physical inactivity [29,90,91], etc.

Gut microbiota affects the pathogenesis of GSD through several mechanisms. Some bacteria alter the composition of bile directly via *β*-glucuronidase, cholyl-glycyl hydrolase, phospholipase A1, or urease activities, or by biofilmformationthereby promoting calcium bilirubinate (pigment) stone generation [92,93]. Till now, it has not been clear whether bacterial pathogens of the biliary tree contribute to the stone formation or alternatively if the presence of gallstones promotes chronic colonization [15]. The activity of the gut microbiota could also be linked to the development of GSD by altering the concentration of serum lipids [94], and biliary lipids in bile and/or increasing the faecal excretion of bile salts [95]. Gut microbiota can modulate bile acid metabolism through the activity of bile salt hydrolases, which deconjugate bile acids, and the activity of 7α-dehydroxylase, which converts primary bile acids (cholic acid and chenodeoxycholic acid) to secondary bile acids (deoxycholic acid and lithocholic acid) [94].

Bile acids regulate metabolism via activation of specific nuclear receptors (e.g., farnesoid X receptor, pregnane X receptor, vitamin D receptor, and cell surface G protein-coupled receptors, such as the G protein-coupled bile acid receptor (TGR5 and Gpbar-1)) [96,97]. The effect of the farnesoid X receptor is antilithogenic:farnesoid X receptor activation in the intestine by bile acids induces fibroblast growth factor 15 expression, which suppresses the expression of cholesterol 7α-dehydroxylase in the liver [98]. Gallstone patients had significantly higher levels of 7α-dehydroxylating bacteria than individuals without gallstones [99]. The increase of 7α-dehydroxylation activity of the intestinal microflora promoted the deoxycholic acid excess in the bile acid pool [100], and the increase in the percentage of deoxycholic acid in bile and bile acid hydrophobicity leads to a decrease in the cholesterol microcrystal nucleation time and the formation of cholesterol gallstones [101].

## 4. The Firmicutes/BacteroidetesRatio and GSD

### 4.1. Gut Microbiota

#### 4.1.1. Gut Microbiota in Mice and Cholelithiasis

Many reports are underlining the association of the gut microbiota with the pathogenesis of cholesterol cholelithogenesis in mice [15,102,103] and humans [5,6,80,81,82,93,100,104,105,106,107,108,109,110,111,112,113,114,115,116,117,118,119].

Alteration of indigenous gut microbiota by bacteria transferring has been shown to make germ-free mice more susceptible to the formation of cholesterol gallstones [102]. In a study of mice without and with cholesterol gallstones (induced by a lithogenic diet) using 16S rRNA gene sequencing, it was found that in the faeces of mice, the *Firmicutes/Bacteroidetes* ratio and the *Firmicutes* content decreased (from 59.71% under chow diet to 31.45% under lithogenic diet, *p* < 0.01), the richness and alpha diversity of the microbiota also significantly reduced [103]. Cholelithogenic enterohepatic *Helicobacter* spp. (phylum *Proteobacteria*) have been identified and their important role in the formation of cholesterol gallstones in mice and perhaps in humans has been shown [15].

#### 4.1.2. Gut Microbiota in Humans and Gallstones

In the gallstone group included 30 patients, the diversity of intestinal bacteria and the abundances of certain phylogroups significantly decreased, especially *Firmicutes*, the *Firmicutes/Bacteroidetes* ratio was also significantly decreased compared with the control group included 30 healthy individuals [6]. 7α-dehydroxylating gut bacteria (the *Clostridium* genus) were significantly increased, whereas cholesterol-lowering bacteria (the *Eubacterium* genus) were significantly reduced. *Clostridium* was positively correlated with secondary bile acids. It can be assumed that an increase in *Clostridium* and a decrease in *Eubacterium* contribute to bile saturation with cholesterol in patients with gallstones [100]. In the gallstone group, *Ruminococcus gnavus* could be used as a biomarker, while in the control group–*Prevotella 9* and *Faecalibacterium* [6].

Keren et al. (2015) showed that intestinal microbial diversity, the abundances of the genus *Roseburia* and the species *Bacteroides uniformis* were decreased, and those of the family *Ruminococcaceae* and the genus *Oscillospira* were increased in patients with gallstones before cholecystectomy compared with the controls [5]. After cholecystectomy in the patients with gallstones, the abundance of the phylum *Bacteroidetes*, and also the family *Bacteroidaceae* and the genus *Bacteroides* showed a significant increase. Gallstone patients had higher overall concentrations of faecal bile acids [5]. *Roseburia* was significantly positively correlated with faecal cholesterol, but not with bile acids; *Oscillospira* correlated negatively with primary bile acids and faecal cholesterol concentration and positively–with the secondary bile lithocholic acid in the faeces. Thus, the authors suggest that *Oscillospira* may predispose individuals to cholesterol gallstones [5]. Cholecystectomy alters bile flow into the intestine and bidirectional interactions between bile acids and intestinal microbiota, thereby increasing bacterial degradation of bile acids into faecal secondary bile acids [104,105]. Deoxycholic acid can inhibit the growth of thececal microbiota in rats; moreover, members of the *Bacteroidetes* phylum (*Bacteroides vulgatus, Bacteroides sartorii*) are more sensitive to secondary bile acids exposure than members of the *Firmicutes* phylum (*Clostridium innocuum*, *Blautia coccoides*) [120]. Deoxycholic acid concentrations were negatively correlated with the *Bacteroidetes* phylum in patients with GSD [5]. Increasing levels of the cholic acid cause a dramatic shift toward the *Firmicutes* (from 54.1% before of administration of cholic acid up to 95% after [120]), particularly *Clostridium* cluster XIVa and increasing production of the harmful deoxycholic acid [104,121].

Wang W et al. (2018) identified ageing-associated faecal microbiota in a healthy population, which was lost in cholecystectomy patients [81]. Absent intestinal bacteria, such as *Bacteroides*, were also negatively related to secondary bile acids in cholecystectomy patients. The abundances of *Prevotella, Desulfovibrio, Barnesiella, Paludibacter*, and *Alistipes* all decreased, whereas those of *Bifidobacterium, Anaerostipes*, and *Dorea* all increased in the cholecystectomy patients [81].

In the frame of a case-control study, Yoon W et al. (2019) showed that *Blautia obeum* and *Veillonella parvula*, which have azoreductase activity, were more abundant in faecal samples in the 27 patients of the cholecystectomy group compared to the control group [82]. The abundance of family * Muribaculaceae* belonging to the phylum *Bacteroidetes* was decreased and that of the family *Lactobacillaceae* was increased in the cholecystectomy group. At the genus level, the abundance of *Ruminococcus* was greater in the cholecystectomy group [82].The actual number of taxa observed in a faecal sample was significantly lower in the cholecystectomy group. However, the difference in the diversity of the gut microbiota between the cholecystectomy and control groups was subtle [82].

Two years after cholecystectomy, eight patients with the symptomatic post-cholecystectomy syndrome, eight patients with the asymptomatic post-cholecystectomy syndrome, and eight healthy individuals were examined [106]. It was shown that *Firmicutes* and *Bacteroidetes* had similar abundance and contents among the three groups. The gut microbiome of the symptomatic post-cholecystectomy syndrome patients was dominated by *Proteobacteria* in faeces and contained little *Firmicutes* and *Bacteroidetes* [106].

Wu et al. (2013) studied the composition of bacterial communities of the gut, bile, and gallstones from 29 cholesterol gallstone patients and the gut of 38 healthy controls [107] by 16S rRNA gene sequencing method. They found a significant increment of the gut bacterial phylum *Proteobacteria* anddecrement of gut bacterial genera *Faecalibacterium*, *Lachnospira*, and *Roseburia*. When compared with gut, a significantly decreased level of the bacterial phylum *Bacteroidetes* in the biliary tract was found. The *Firmicutes/Bacteroidetes* ratio in faeces in patients with GSD did not differ in comparison with the control group [107].

Ren X et al. (2020) examined stool samples from 104 subjects (equally post-cholecystectomy patients and healthy controls) which were collected for 16S rRNA gene sequencing to analyze the bacterial profile [80]. It was shown noteworthy compositional and abundant alterations of bacterial microbiota in post-cholecystectomy patients, characterized as *Bacteroides ovatus, Prevotella copri*, and *Fusobacterium varium* remarkably increased; *Faecalibacterium prausnitzii, Roseburia faecis*, and *Bifidobacterium adolescentis* significantly decreased. Machine learning-based analysis, that integrates gut microbiota and other anthropometric parameters, showed a pivotal role of *Megamonas funiformis* in discriminating post-cholecystectomy patients from healthy controls. Additionally, the duration after cholecystectomy notably affected bacterial composition in post-cholecystectomy patients [80].

Eventually, if we summarize the results of most studies of the microbiota in patients with GSD different authors found both a significant increment of gut bacterial phyla *Firmicutes* (*Lactobacillaceae* family, genera *Clostridium*, *Ruminococcus*, *Veillonella*, *Blautia*, *Dorea, Anaerostipes*, and *Oscillospira*), *Actinobacteria* (*Bifidobacterium* genus), *Proteobacteria*, *Bacteroidetes* (genera *Bacteroides*, *Prevotella*, and *Fusobacterium*) (Figure 1) and significant decrement of gut bacterial phyla *Bacteroidetes* (*Muribaculaceae* family, and *genera Bacteroides, Prevotella, Alistipes, Paludibacter,*
*Barnesiella)*, *Firmicutes* (genera *Faecalibacterium*, *Eubacterium*, *Lachnospira*, and *Roseburia*), *Actinobacteria* (*Bifidobacterium* genus), and *Proteobacteria* (*Desulfovibrio* genus) (Figure 2). In other words, in patients with GSD, an increase and decrease in almost all major intestinal bacterial phyla were detected. In one study the *Firmicutes/Bacteroidetes* ratio in faeces in patients with GSD was significantly decreased in comparison with the controls [6], in two studies–did not differ [106,107]. In addition to *Firmicutes* and *Bacteroidetes* as the main phyla, *Proteobacteria* and other phyla may contribute to the gut dysbiosis in patients with GSD.

Using metagenomic DNA sequencing, researchers have been able to categorize individuals as either high gene count (HGC) or low gene count (LGC) [44]. HGC individuals are generally considered to have a greater repertoire of microbial metabolic functions, a functionally more robust gut microbiome, and greater overall health, including a lower prevalence of obesity and metabolic disorders [48]. Examples of bacterial taxa that have been associated with human health and proper gastrointestinal function include *Bacteroides, Bifidobacterium, Clostridium* clusters XIVa and IVa (butyrate producers), *Eubacterium, Faecalibacterium, Lactobacillus,* and *Roseburia*. Bacterial species that might protect against weight gain and are enriched in HGC individuals include *Anaerotruncus colihominis, Butyrovibrio crossotus, Akkermansia* spp., and *Faecalibacterium* spp. [48]. The studies of the gut microbiota in patients with GSD included in our review demonstrated a reduction of bacterial taxa that have been associated with human health, i.e., genera *Bacteroides, Faecalibacterium*, *Roseburia*, *Eubacterium*, an increase in *Lactobacillaceae* family, and oppositely directed changes in *Bifidobacterium*.

#### 4.1.3. Bile Microbiota in Humans and Gallstones

The presence of bacterial amplicons belonging to *Firmicutes, Bacteroidetes*, and *Actinobacteria,* and *Proteobacteria* phyla in the human intact gallbladder bile was proved by 16S rRNA gene sequencing [108,109]. Associations between alpha- and beta-diversity, a taxonomic profile of bile microbiota (*Bacteroidetes, Proteobacteria, Actinobacteria,* and *Firmicutes* phyla, analyzed with 16S rRNA gene sequencing), and taurocholic and taurochenodeoxycholic bile acid levels were evidenced in 37 Russian patients with GSD [110].

At the phylum level, *Bacteroidetes* was statistically more abundant in the bile of patients with GSD (24.00%) compared to the control (13.49%) [109]. Members of the families *Bacteroidaceae*, *Prevotellaceae*, *Porphyromonadaceae*, and *Veillonellaceae* were more frequently detected in patients with GSD. The genus *Dialister* and enterobacteria *Escherichia-Shigella* also showed a significantly higher representation in the bile in the patients with GSD [109]. The Shannon diversity index was statistically higher in the bile of the control group than that obtained in the patients with GSD [102].However, it was not taken into account that bile samples from the gallbladder of individuals from a control group were obtained from liver donors, and they were not only treated with antibiotics but also not fully examined to exclude hepatobiliary or other important pathology [109].

The *Proteobacteria*, *Firmicutes*, *Bacteroidetes,* and *Actinobacteria* phyla dominated the biliary microbiota in the persons, all of whom were diagnosed with GSD, at that biliary tract microbiota of participants with GSD showed substantial person-to-person variation [79]. Metagenomic sequencing of bile from gallstone patients showed that oral cavity/respiratory tract inhabitants were more prevalent than intestinal inhabitants [108]. At the same time, bile samples from gallstone patients had reduced microbial diversity compared to healthy faecal samples [108]. Among patients with the new onset of common bile duct stones, five dominant phyla were identified: *Proteobacteria* (60%), *Firmicutes* (27%), *Bacteroidetes* (4%), *Actinobacteria* (3%), and Unclassified_Bacteria (3%) in biliary microbiota [111]. At the genus level, the five genera with the highest relative abundances in patients with the new onset of common bile duct stones were *Escherichia/Shigella*, *Halomonas*, *Klebsiella, Streptococcus,* and *Enterococcus* [111].

In patients with cholangiolithiasis associated with sphincter of Oddi laxity, *Proteobacteria* and *Firmicutes* were the most widespread phylotypes, especially *Enterobacteriaceae*, in the bile, which was collected intraoperatively [112]. In the bile of the cholecystectomized gallstone patients *Escherichia coli*, *Salmonella* sp., and *Enterococcusgallinarum* were detected by using next-generation sequencing technology [113]. *Enterobacteriaceae* are frequently isolated from bile aspirates or gallbladder bile from GSD patients using cultural [114,115] and culture-independent techniques [116,117]. The biliary microbiota (investigated by using 16S rRNA amplicon sequencing) had a reduced diversity comparatively with the duodenal microbiota in gallstone patients [117]. Although the majority of identified bacteria were greatly diminished in bile samples, three *Enterobacteriaceae* genera (*Escherichia*, *Klebsiella*, and an Unclassified genus) and *Pyramidobacter* were abundant in bile [117].

In terms of bile microbial distribution, analyzed by the 16S rRNA encoding gene (V3-V4), patients with recurrent common bile duct stone had significantly higher *Proteobacteria*, while *Bacteroidetes* and *Actinobacteria* are significantly lower compared with the control group at the phylum level [117]. At the family level, *Enterobacteriaceae* was significantly abundant in the bile samples of the recurrence stone group compared with the control group. At the genus level, the recurrence stone group had significantly more *Escherichia*. The diversity of bile microbiome in patients with recurrent common bile duct stone is lower than that in the control non-cholelithiasis group [117].

During a cholecystectomy, mucosal DNA extraction and metagenomic sequencing were performed to evaluate changes in the microbiota between chronic calculous cholecystitis and gallbladder cancer patients [118]. At the phylum level, *Firmicutes*, *Bacteroidetes*, *Actinobacteria*, and *Proteobacteria* were found to be stable in both groups. The diversity of the biliary microbiota was significantly lower in the calculous cholecystitis group, compared with the gallbladder cancer group [118].

In four patients who underwent cholecystectomy for acute calculous cholecystitis metagenome analysis of bile, faeces, and saliva was performed [119]. In all the examined patients with acute calculous cholecystitis, *Escherichia coli* (*Enterobacteriaceae* family) was found in large quantities in the bile, in two of them-also in the faeces, in the third patient, *Bifidobacterium* prevailed in the faeces. This is not enough to conclude the relationship between the intestinal microbiota and acute calculous cholecystitis, since if bile samples were taken during surgery, then saliva and faeces were collected by patients during hospitalization (it is not clear before or after the cholecystectomy) [119].

During endoscopic retrograde cholangiopancreatography, a total of 44 bile samples of patients with GSD were collected. Bacterial infection in bile samples was detected in 54.5% of patients with GSD. *Escherichia coli* showed a significant association with gallstones [122].

Thus, bile samples from patients with GSD had reduced microbial diversity in some studies and increased microbial diversity in others compared to healthy faecal samples. Nevertheless, most authors recognize that patients with GSD have reduced bacterial diversity of intestinal and bile microbiota. The phyla *Proteobacteria* (especially family *Enterobacteriaceae*) and *Firmicutes* (*Enterococcus* genus) were more often detected in the bile of patients with GSD, and the phyla *Bacteroidetes* and *Synergistes* (*Pyramidobacter* genus) were less frequently detected.

Some reports described live bacteria and bacterial DNA as long-term constituents in different fat depots in obesity and diabetes mellitus type 2 [123,124]. In humans with the metabolic syndrome, altered microbiome composition together with a defective intestinal barrier has been suggested to facilitate translocation of microbes, thereby contributing to low-grade inflammation. A recent study demonstrated a bacterial signature in mesenteric adipose tissues without the obvious presence of blood: members of the *Enterobacteriaceae* family compartmentalize in the extra-intestinal tissues of people with diabetes mellitus type 2 independently of obesity [123]. The authors suggest that members of the *Enterobacteriaceae* family are key players in diet-induced dysmetabolism in the host. Unfortunately, the intriguing topic of possible translocation of living bacteria (perhaps even members of the *Enterobacteriaceae* family) from the gut to other body sites in patients with GSD remains undiscovered.

So, when analyzing available studies of intestinal and bile microbiota in animals and patients with GSD [5,6,15,79,80,81,82,92,93,100,102,103,104,105,106,107,108,109,110,111,112,113,114,115,116,117,118,119] there were no unidirectional changes in the *Firmicutes/Bacteroidetes* ratio. This situation with opposite results is typical not only for GSD. For comparison, we will briefly present the results of several studies reporting differences in phylum levels in patients with non-alcoholic fatty liver disease (usually associated with obesity): the phylum *Bacteroidetes*–increased [86], decreased [88,125], did not differ [87,126], the phylum *Firmicutes*–decreased [86,87], increased [126], and the *Firmicutes*/*Bacteroidetes* ratio decreased [88].

This variation in the relative abundance of the phylum of the gut corresponds to the analysis of seven studies in Finucane et al. (2014): *Bacteroidetes*–from 0% to 90%, *Firmicutes*–from 0 to 100% [63]. This also applies to GSD. For example, the highest abundance of *Firmicutes* phylum in the human gastrointestinal tract in one GSD patient was 93.30% and the lowest was 1.17% in another. A similar result was also seen in bile with a high of 55.10% and low of 0.08% [107]. In another study, the range of relative abundance of *Firmicutes* phylum was 0–92% in the bile of patients with GSD [79].

## 5. Some Reasons for the Lack of Unity in the Assessment of the *Firmicutes/Bacteroidetes* Ratio

Gut microbiota is changing with human development and is influenced by many confounding variables which could prevent the existence of a unique taxonomic signature as a standard feature for obesity and associated comorbidities such as GSD [64,83,89].

Gender, age, differences in host genetics [4]. There are differences in the gut microbiota between males and females, such as higher levels of ***Bacteroides–Prevotella*** group in males [127] and a higher proportion of *Firmicutes* in females [128]. However, Bezek et al. (2020) found the highest abundance of *Bacteroidetes* phylum in females [51]. The *Firmicutes/Bacteroidetes* ratio evolves during different life stages. For infants (up to 10 months), adults (25–45 years), and elderly individuals (70–90 years), these ratios were 0.4, 10.9, and 0.6, respectively [44].Vaginal delivery or C-section, methods of milk feeding [129].Changes in the gut microbiota under the influence of a variety of diets have been widely studied [30,31,32,35,36,38,47,52,62,72,73,84,91,129,130,131]. It was noted that the amount of stool energy in a proportion of ingested calories was positively correlated with the abundance of the phylum *Bacteroidetes* and negatively–with the abundance of the phylum *Firmicutes* in the faeces [38]. As a rule, the “western diet” increases biliary secretion of bile acids and reshapes the gut microbiota in obesity by increasing the *Firmicutes* and decreasing the *Bacteroidetes* [35,62]. Several population-based studies have shown that populations given increased amounts of polyunsaturated fats have a significant risk of developing gallstones [9,12,132,133,134]. The MICOL study, however, showed no such association [135]. Gutiérrez-Díaz et al. (2018) support a link between diet, biliary microbiota, and GSD [84]. Comparing to health control in patients with GSD, dairy product intake was negatively associated with the proportions of *Bacteroidaceae* and *Bacteroides*, and several types of fibre, phenolics, and fatty acids were linkedto the abundance of *Bacteroidaceae, Chitinophagaceae, Propionibacteraceae, Bacteroides,* and *Escherichia‒Shigella* [84]. However, the timing of these changes is surprising. In response to dietary perturbations, the gut microbiota took from 24 h [130] to 3.5 days [36] to change detectably and reaches a new steady state. Repeated dietary shifts demonstrated that most changes to the gut microbiota are reversible [36]. Also, Carmody et al. (2015) suggest, that the effects of dietary intake overshadow any pre-existing differences between strains due to host genotype [36]. Add to this the inter-individual variability in the processing of dietary compounds by the human gastrointestinal tract [136] and the hope of finding patterns in the relationship “microbiota–host–diet” becomes quite vague.The presence of pathological conditions (diabetes mellitus [83], cardiovascular disease [89], inflammatory bowel disease [64], etc.). For example, the sphincter of Oddi laxity is associated with cholangiolithiasis, probably due to enhanced reflux of intestinal contents that change the microenvironment [112]. Compared with patientswith cholangiolithiasis with normal sphincter of Oddi, patients with sphincter of Oddi laxity possessed more varied microbiota [112].Treatment (antibiotics [137], metformin [138], etc.). Metformin shifts gut microbiota composition through the enrichment of *Akkermansia muciniphila* as well as several SCFA-producing microbiotas (*Butyrivibrio, Bifidobacterium bifidum,* etc.) [138].The influence of exercise training on the gut microbiome has also been examined [29,91,131] and it has been shown that exercise alone increased the *Firmicutes/Bacteroidetes* ratio, irrespective of diet [91].Human microbiota differs according to the geographical location of the studies [113,139,140,141]. It was found a positive correlation between *Firmicutes* and latitude and a negative correlation between *Bacteroidetes* and latitude [139]. In the frame of study of human gut microbiota community structures in urban and rural populations in Russia, two clusters were obtained: the first was driven by the genus *Prevotella*, and the second exhibited a high representation of *Bifidobacterium* and various genera of the phylum *Firmicutes*. The urban and rural metagenomes were distributed equally between the clusters: 53% of the first and 52% of the second cluster were urban [141].Lifestyle. Sleep deprivation correlates with changes in the gut microbiome, with an increase of the *Firmicutes/Bacteroidetes* ratio, higher abundances of the families *Coriobacteriaceae* and *Erysipelotrichaceae*, and lower abundance of *Tenericutes* [51,142]. Stress, occupation, temporal dynamics and stability of the microbiome: diurnal oscillations in the relative abundance of almost 10% of all bacterial taxawere detected [143].The extreme variability of the *Firmicutes/Bacteroidetes* ratio can be attributed to the different experimental designs (insufficient sample size [144]), microbiota fingerprinting, and genome analyses (choice of the primers the 16S rRNA target region, DNA extraction technique [145], and sequencing platform) [50,146]. Besides, members of the *Bacteroidetes* and *Actinobacteria* were significantly more stable components of the microbiota than the population average, while the *Firmicutes* and *Proteobacteria* were significantly less stable [147]. The plasticity vs. stability dichotomy of the human microbiome was confirmed in a study by David et al. (2014): when analyzing microbiota samples over several months, only 5% of the gut species were defined as belonging to a stable temporal core microbiome. Yet, each person still maintained a personalized microbiome [140].There are also hard-to-determine factors, such as the Earth’s geomagnetic field, weather, etc.

## 6. Conclusions

Meta-analysis has shown that the microbial changes associated with obesity may be minor shifts in the community that escape detection with significance tests [77]. It may be the case that the microbiome’s effect on obesity is not mediated through its taxonomic composition but rather its function, since closely related taxa can have widely varying functions and distantly related taxa can have similar functions [63]. It is proved that variable combinations of species from different phyla could ‘presumptively’ fulfil overlapping and/or complementary functional roles required by the host, a scenario where minor bacterial taxa seem to be significant active contributors [39]. For example, the cocolonization of germ-free mice with *B. Thetaiotaomicron* and *E. rectale* constitutes a mutualism, in which both members show a clear benefit [148] and the efficiency of fermentation of dietary polysaccharides to short-chain fatty acids by *B. thetaiotaomicron* increases in the presence of *M. Smithii* [149].

Based on the analysis of the great number of contradictory results reported in the literature, it is currently difficult to associate specific microbial signatures or the *Firmicutes/Bacteroidetes* ratio with determining health status and more specifically to consider it as a hallmark of GSD and/or obesity. However, most authors believe that both obesity [33,34,40,48,64,130] and GSD [5,6,81,103,109,111,117,118,119,120] are associated with reduced microbial diversity.Therefore, it is important to look at the overall composition of the gut microbial population structure as an indicator of obesity and obesity-associated pathologies, such as GSD, rather than simply the *Firmicutes/Bacteroidetes* ratio [150]. However, in my opinion, it is possible to modify this ratio, e.g., to introduce a coefficient that characterizes BMI, to calculate the ratio not of the *Firmicutes* phylum, but only of the *Clostridia* class, and so on.

Further studies should focus on the possibility of modulating the intestinal microbiota to find out whether variations in the microbiota may be a target for lowering the risks and prevalence rates of GSD. Future studies to identify specific bacterial species or populations associated with the obesity or GSD phenotype will help optimize disease therapies through microbiome-informed patient stratification, through personalized treatment decisions. A better understanding of bacterial communities in both the gut and biliary tract of gallstone patients is crucial in developing strategies to promote personalized microbiome-based GSD prediction and treatment responsiveness.

## Figures and Tables

**Figure 1 jpm-11-00013-f001:**
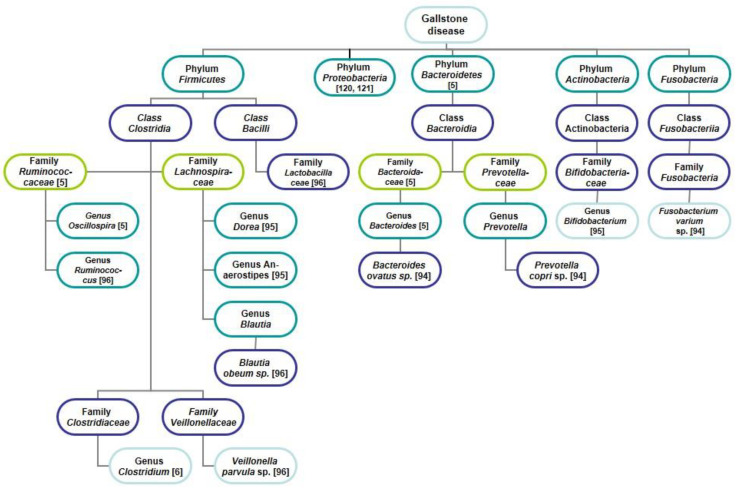
Characteristics of the gut microbiome of patients with GSD. A significant increase of the phyla *Firmicutes*, *Actinobacteria*, *Proteobacteria*, and *Bacteroidetes* is reflected. The number in square brackets indicates a reference in the list of references.

**Figure 2 jpm-11-00013-f002:**
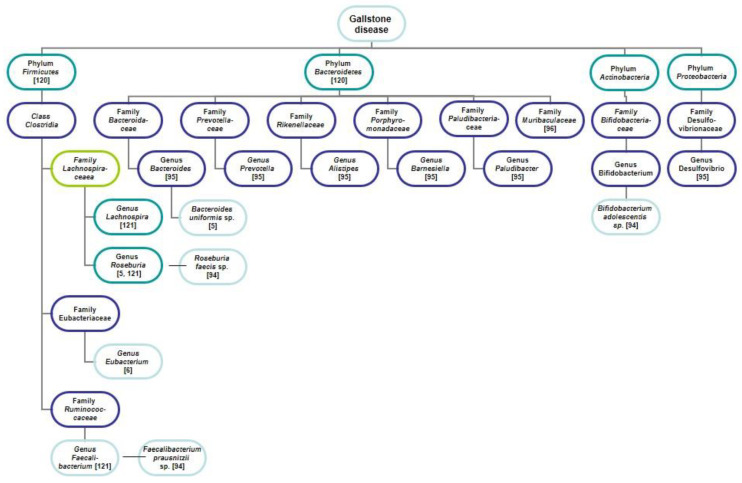
Characteristics of the gut microbiome of patients with GSD. A significant decrease of the phyla *Firmicutes*, *Actinobacteria*, *Proteobacteria*, and *Bacteroidetes* is reflected. The number in square brackets indicates a reference in the list of references.

## Data Availability

Not applicable.

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
