# Peer review of "Gallstone Disease, Obesity and the Firmicutes/Bacteroidetes Ratio as a Possible Biomarker of Gut Dysbiosis"

_jpm, 2020, doi:10.3390/jpm11010013_

Round 1

Reviewer 1 Report

Author Grigor’eva, have performed a thorough narrative review of literation on bi-direction relation, changes, and implication between Gall stone disease, obesity and microbiota. This paper covers a wide breadth of literature, significantly points out and summarizes the taxonomic changes associated during GSD and obesity. Importantly, the opportunistic pathogens which may play a role during the disease are shown to increase. The author also highlights the importance of the two major bacterial phyla Firmicutes and Bacteroidetes, and the F/B ratio during obesity. F/B ratio has been shown as an indirect indicator of disease/inflammation during various diseases including obesity. This is a well-written paper with important caveats and highlights during the GSD and obesity. Although, this is not a meta-analysis or systematic review, the paper as a narrative review is well-written. However, a meta-analysis of literature would have given as statistical picture of the whole microbiota changes and its implications per se. A few points I would like the author to include in the current work: 1. Most of the papers/literature discussed in the paper are related to fecal microbiome, perhaps it should be mentioned. Moreover, the recent papers by Massier et al. Gut journal and Anhe et al. Nature metabolism are two major breakthroughs paper in the field of obesity/microbiome. These papers are the first showcase the presence of microbes-bacteria in the adipose tissue of obese/unhealthy volunteers. Adding a section on this will be beneficial. 2. The role of microbial translocation leading to unhealthy conditions should be highlighted if there is a chance to include. Because fecal microbiome is difference from tissue or the actual mucosal microbiome. 3. Included a table of the studies can be beneficial but not necessary 4. Perhaps a figure/illustration may add value too.

Author Response

Dear Reviewer,

I thank you for your kind attention to my manuscript and for your friendly constructive criticism. Your contribution will definitely help improve my manuscript.

Responses to Your comments.

  1. Most of the papers/literature discussed in the paper are related to fecal microbiome, perhaps it should be mentioned. Moreover, the recent papers by Massier et al. Gut journal and Anhe et al. Nature metabolism are two major breakthroughs paper in the field of obesity/microbiome. These papers are the first showcase the presence of microbes-bacteria in the adipose tissue of obese/unhealthy volunteers. Adding a section on this will be beneficial.
  1. The role of microbial translocation leading to unhealthy conditions should be highlighted if there is a chance to include. Because fecal microbiome is difference from tissue or the actual mucosal microbiome.

I agree with Your comment and propose a new text: “Some reports described live bacteria and bacterial DNA as long-term constituents in different fat depots in obesity and diabetes mellitus type 2 [116 Anhe, 117 Massier]. In humans with the metabolic syndrome, altered microbiome composition together with a defective intestinal barrier has been suggested to facilitate translocation of microbes, thereby contributing to low-grade inflammation. a recent study demonstrated a bacterial signature in mesenteric adipose tissues without the obvious presence of blood: members of the Enterobacteriaceae family compartmentalize in the extra-intestinal tissues of people with diabetes mellitus type 2 independently of obesity [116 Anhe]. The authors suggest that members of the Enterobacteriaceae family are key players in diet-induced dysmetabolism in the host. Unfortunately, the intriguing topic of possible translocation of living bacteria (perhaps even members of the Enterobacteriaceae family) from the gut to other body sites in patients with GSD remains undiscovered”. Rows 376-386.

  1. Included a table of the studies can be beneficial but not necessary
  2. Perhaps a figure/illustration may add value too.

I agree with Your comment and added two figures/illustrations.

Dear Reviewer,

I look forward to hearing from you regarding the submission of my manuscript. I wish you good health.

Sincerely yours, Irina Grigor’eva.

Reviewer 2 Report

Manuscript ID: jpm-1019881

Title: Gallstone disease, obesity and the Firmicutes/Bacteriodetes ratio.

Authors: Irina N. Grigor’eva

The above manuscript may be potentially interesting. The author has collected a large number of actual articles on the topic presented in the manuscript. However, it should be noted, however, that the language of the manuscript, the form of presentation and interpretation of the collected data are unacceptable. Hardly any sentence can be accepted as it stands. For this reasons, the current version of the manuscript cannot be accepted. Apart from the linguistic problems, the presence of numerous substantive and interpretation errors suggests that the authors is not a medical doctor. The author should cooperate with a mentor who would assist in the redaction of the new version of the manuscript. It may be, for example, a fluent English-speaking experienced gastroenterologist from the Department of Gastroenterology of the Medical University in the author’s city residence. Such cooperation could be fruitful for both partners.

Below there are examples of some errors found in the manuscript:

  1. The title of the manuscript. The author should specify where the relation of Firmicutes to Bacteroidetes was determined.
  2. Abstract, the first sentence. “Both obesity and microflora contribute to the formation of gallstones, but information is fragmented” What kind of microflora and where was it observed? Moreover, the word “fragmented” should be replaced by “fragmentary” or “incomplete”.
  3. Abstract, the second sentence. “Increased the Firmicutes/Bacteroidetes ratio is usually observed with obesity, but other trials reported even the opposite”. “with obesity” should be replaced by “ “in obesity” or “in patients with obesity”. The second part of the sentence “but other trials reported even the opposite” could be replaced by “but some studies show the opposite relationship”.
  4. Abstract, the third sentence: “The aim of this review was to discuss the role of the Firmicutes/Bacteroidetes ratio as a possible marker of gallstone disease (GSD)…” There are numerous cheaper and more specific methods for detection of GSD. For this reason, the author should change the aim of the study.
  5. Abstract, the last sentence. “remain disputable” should be replaced by “in the gut remain unclear” or something similar.
  6. The body of the manuscript. The data presentation is chaotic. The author should clearly state where the bacterial flora was tested present the results in separate sections depending on the form and place of collection of the material.
  7. Data obtained in patients and in experiments with animals should be presented separately.
  8. During presenting the microflora studies, the author should present the method use in these studies. In contrast to bacterial culture, bacterial DNA extraction followed by amplicon sequencing targeting hypervariable region of the 16S rRNA gene have some limitation. Extracted DNA mainly comes from live bacteria, but this method also isolates DNA from dead bacteria, as well as from bacterial residues. For this reason, obtained results may be more dependent on the bacteria present in the diet than on bacteria living in subsequent sections of gastrointestinal tract
  9. All abbreviations should be given in their full name in the place of the first use.

Author Response

Dear Reviewer,

Thank you for your careful reading and comments. I appreciate the qualified assessment and information assistance You provided, and Your contribution will definitely help improve my manuscript.

Responses to Your comments.

However, it should be noted, however, that the language of the manuscript, the form of presentation and interpretation of the collected data are unacceptable. Hardly any sentence can be accepted as it stands. For this reasons, the current version of the manuscript cannot be accepted. Apart from the linguistic problems, the presence of numerous substantive and interpretation errors suggests that the authors is not a medical doctor. The author should cooperate with a mentor who would assist in the redaction of the new version of the manuscript. It may be, for example, a fluent English-speaking experienced gastroenterologist from the Department of Gastroenterology of the Medical University in the author’s city residence. Such cooperation could be fruitful for both partners.

I changed the presentation form (added two figures) and data interpretation. I almost completely rewrote several sections of the article. The language of the article was partially improved with the help of the Grammarly program, but I plan to use the linguistic help of the Editorial Board for a fee and solve this problem. I am a medical doctor and we don't have a specialist on such a narrow topic and I can't get advice from colleagues. 

  1. The title of the manuscript. The author should specify where the relation of Firmicutes to Bacteroidetes was determined.

I changed the title of the article according to Your instructions. “Gallstone disease, obesity and the Firmicutes/Bacteriodetes ratio AS A POSSIBLE BIOMARKER OF GUT DYSBIOSIS”.

  1. Abstract, the first sentence. “Both obesity and microflora contribute to the formation of gallstones, but information is fragmented” What kind of microflora and where was it observed? Moreover, the word “fragmented” should be replaced by “fragmentary” or “incomplete”.

I agree with Your comment and I deleted this sentence.

  1. Abstract, the second sentence. “Increased the Firmicutes/Bacteroidetes ratio is usually observed with obesity, but other trials reported even the opposite”. “with obesity” should be replaced by “ “in obesity” or “in patients with obesity”. The second part of the sentence “but other trials reported even the opposite” could be replaced by “but some studies show the opposite relationship”.

I agree with Your comment and I deleted this sentence.

  1. Abstract, the third sentence: “The aim of this review was to discuss the role of the Firmicutes/Bacteroidetes ratio as a possible marker of gallstone disease (GSD)…” There are numerous cheaper and more specific methods for detection of GSD. For this reason, the author should change the aim of the study.

I agree with Your comment and I changed this sentence according to Your instructions:” we provide an overview of the current evidence on the composition of the gut and the bile microbiota in patients with GSD with reference to the Firmicutes/Bacteroidetes ratio, as a possible biomarker of obesity, given that obesity is a key risk factor for GSD.”

  1. Abstract, the last sentence. “remain disputable” should be replaced by “in the gut remain unclear” or something similar.

I agree with Your comment and I changed this sentence according to Your instructions: “The results of the reviewed studies on the relationship between GSD, obesity, and the Firmicutes/Bacteroidetes ratio remain unclear”. Row 25.

  1. The body of the manuscript. The data presentation is chaotic. The author should clearly state where the bacterial flora was tested present the results in separate sections depending on the form and place of collection of the material.

I agree with Your comment and I changed the body of the manuscript. I redesigned the article structure, I deleted rows: 67-68, 72-84, 92-105, 128-149,156-175, 404-408, 427-428, 438-439,446, 447, 449, 454-460, 462-463, 469-474, and I added rows: 34-35, 39-63, 72-74, 97-105, 112-117, 123-124, 205-207, 297-299, 316-318. I removed 32 references and added three references. I specified the place of the material collection.

  1. Data obtained in patients and in experiments with animals should be presented separately.

I agree with Your comment and I presented separately data obtained in patients and in experiments with animals: 4.1.1. gut microbiota in mice and cholelithiasis. 4.1.2. Gut microbiota in humans and gallstones. Rows 192 and 206.

  1. During presenting the microflora studies, the author should present the method use in these studies. In contrast to bacterial culture, bacterial DNA extraction followed by amplicon sequencing targeting hypervariable region of the 16S rRNA gene have some limitation. Extracted DNA mainly comes from live bacteria, but this method also isolates DNA from dead bacteria, as well as from bacterial residues. For this reason, obtained results may be more dependent on the bacteria present in the diet than on bacteria living in subsequent sections of gastrointestinal tract

I agree with Your comment and propose a new text: “The extreme variability of the Firmicutes/Bacteroidetes ratio can be attributed to the different experimental designs (insufficient sample size [137]), microbiota fingerprinting, and genome analyses (choice of the primers the 16S rRNA target region, DNA extraction technique [138], and sequencing platform) [41, 139]. Rows 455-458.

  1. All abbreviations should be given in their full name in the place of the first use.

I agree with Your comment and propose a new text: “the World Health Organization (WHO)” and I verified the other abbreviations. Rows 32-33.

Dear Reviewer,

I look forward to hearing from you regarding the submission of my manuscript. I wish you good health.

Sincerely yours, Irina Grigor’eva.

Round 2

Reviewer 2 Report

Manuscript ID: jpm-1019881

Title: Gallstone disease, obesity and the Firmicutes/Bacteriodetes ratio as a possible biomarker of gut dysbiosis.

Authors: Irina N. Grigor’eva

The new version of the manuscript exhibits significant improvement, but there are still some errors that should be corrected before publication.

List of errors:

  1. The abstract is still chaotic. The potential reader starts reading the manuscript from the title and abstract. Therefore, these two elements of the manuscript are particularly important and must be legible. If they are incomprehensible, the potential reader will lose interest in the article. Sentence should be short and understandable. At the beginning of the authors should write about the relationship between obesity and gallstone disease (GSD), as well as between obesity and gut dysbiosis. Then, the author should write that these findings suggest that gallstone disease could be associated with gut dysbiosis. The next sentence should inform that this review presents and summarize the recent findings of studies on the gut microbiota and GSD. The authors should present main findings and the differences between the research results. And finally, the author should write an appropriate conclusion. A new version of the abstract could be, for example, similar to a following form: “Obesity is a major risk factor for developing gallstone disease (GSD). Previous studies have shown that obesity is associated with an elevated ration of the Furmicutes to Bacteroides in in the gut microbiota. These findings suggest that the development of GSD may be related to gut dysbiosis. This review presents and summarizes the recent findings of studies on the gut microbiota in patients with GSD. Most of the studies on the gut microbiota in patients with GSD have shown a significant increase the phyla Firmicutes (Lactobacillaceae family, genera Clostridium, Ruminococcus, Veillonella, Blautia, Dorea, Anaerostipes, and Oscillospira), Actinobacteria (Bifidobacterium genus), Proteobacteria, Bacteroidetes (genera Bacteroides, Prevotella, and Fusobacterium) and a significant decrease in the phyla Bacteroidetes (family Muribaculaceae, and genera Bacteroides, Prevotella, Alistipes, Paludibacter, Barnesiella), Firmicutes (genera Faecalibacterium, Eubacterium, Lachnospira, and Roseburia), Actinobacteria (Bifidobacterium genus), Proteobacteria (Desulfovibrio genus). The influence of GSD on microbial diversity is not clear. Some studies report that GSD reduces microbial diversity in the bile; whereas others suggest the increase in microbial diversity in the bile of patients with GSD. The phyla Proteobacteria (especially family Enterobacteriaceae) and Firmicutes (Enterococcus genus) are most commonly detected in the bile of patients with GSD. On the other hand, the composition of bile microbiota in patients with GSD shows considerable inter-individual variability. The impact of GSD on the ration of the Furmicutes to Bacteroides is unclear and reports are contradictory. For this reason, it should be stated that the results of reviewed studies do not allow for drawing unequivocal conclusions regarding the relationship between GSD and the Firmicutes/Bacteroidetes ratio in the microbiota.
  2. Page 1, Introduction. The author should write a few sentences on the role of gastrointestinal hormones, including ghrelin, in the regulation of food intake and obesity development, as well as protective and healing effects of ghrelin in the gut. For this purpose, the author could use the following interesting articles: Ceranowicz et al. (PMID: 25716961), Cui et al. (PMID: 28232667), Warzecha et al. (PMID: 22300084), Dembinski et al. (PMID: 14726611), Bukowczan et al. (PMID: 25594510), Matuszyk et al. (PMID: 27598133).
  3. The author should again carefully read the manuscript and try to correct still existing errors. For example, page 1 lines 34-35. The abbreviation “etc” can be used when presenting items or concepts belonging to one group with common distinct features. “Etc” means “and so on”. What are the common features of atherosclerosis, diabetes, and gallstone disease, and what other diseases belongs to the same group? The author should add “or” after diabetes and remove “etc”.
  4. Page 1, line 37: The author wrote: “Risk factors of the GSD is age...” What kind of age? Young age, average age or old age? Moreover, in the case of risk factors “is” should be replaced by “are”.
  5. Page 3, line 83. The author wrote: “Other important functions of Bacteroides spp. include the growth of mucus”. Did the author want to write about mucus or mucosa? What does growth of mucus mean? Did the author want to write “the stimulation of mucus production or release” or stimulation of mucosa growth’?
  6. Page 4, lines163-166. The sentence is too long and unclear. This must be corrected, and the some comments about “ect” abbreviation as in the comment 3.

Author Response

Dear Rewer,

Thank you for your invaluable help and comments. You correctly, professionally and clearly presented your recommendations for clarifying some fragments of my manuscript, which will certainly help me.

Responses to Your comments.

  1. The abstract is still chaotic. The potential reader starts reading the manuscript from the title and abstract. Therefore, these two elements of the manuscript are particularly important and must be legible. If they are incomprehensible, the potential reader will lose interest in the article. Sentence should be short and understandable. At the beginning of the authors should write about the relationship between obesity and gallstone disease (GSD), as well as between obesity and gut dysbiosis. Then, the author should write that these findings suggest that gallstone disease could be associated with gut dysbiosis. The next sentence should inform that this review presents and summarize the recent findings of studies on the gut microbiota and GSD. The authors should present main findings and the differences between the research results. And finally, the author should write an appropriate conclusion. A new version of the abstract could be, for example, similar to a following form: “Obesity is a major risk factor for developing gallstone disease (GSD). Previous studies have shown that obesity is associated with an elevated ration of the Furmicutes to Bacteroides in in the gut microbiota. These findings suggest that the development of GSD may be related to gut dysbiosis. This review presents and summarizes the recent findings of studies on the gut microbiota in patients with GSD. Most of the studies on the gut microbiota in patients with GSD have shown a significant increase the phyla Firmicutes (Lactobacillaceae family, genera Clostridium, Ruminococcus, Veillonella, Blautia, Dorea, Anaerostipes, and Oscillospira), Actinobacteria (Bifidobacterium genus), Proteobacteria, Bacteroidetes (genera Bacteroides, Prevotella, and Fusobacterium) and a significant decrease in the phyla Bacteroidetes (family Muribaculaceae, and genera Bacteroides, Prevotella, Alistipes, Paludibacter, Barnesiella), Firmicutes (genera Faecalibacterium, Eubacterium, Lachnospira, and Roseburia), Actinobacteria (Bifidobacterium genus), Proteobacteria (Desulfovibrio genus). The influence of GSD on microbial diversity is not clear. Some studies report that GSD reduces microbial diversity in the bile; whereas others suggest the increase in microbial diversity in the bile of patients with GSD. The phyla Proteobacteria (especially family Enterobacteriaceae) and Firmicutes (Enterococcus genus) are most commonly detected in the bile of patients with GSD. On the other hand, the composition of bile microbiota in patients with GSD shows considerable inter-individual variability. The impact of GSD on the ration of the Furmicutes to Bacteroides is unclear and reports are contradictory. For this reason, it should be stated that the results of reviewed studies do not allow for drawing unequivocal conclusions regarding the relationship between GSD and the Firmicutes/Bacteroidetes ratio in the microbiota.

response. I agree with Your comment and I accept your suggestion about the abstract text with replacement: Furmicutes to Bacteroides to Firmicutes/Bacteroidetes.

Obesity is a major risk factor for developing gallstone disease (GSD). Previous studies have shown that obesity is associated with an elevated the Firmicutes/Bacteroidetes ratio in the gut microbiota. These findings suggest that the development of GSD may be related to gut dysbiosis. This review presents and summarizes the recent findings of studies on the gut microbiota in patients with GSD. Most of the studies on the gut microbiota in patients with GSD have shown a significant increase in the phyla Firmicutes (Lactobacillaceae family, genera Clostridium, Ruminococcus, Veillonella, Blautia, Dorea, Anaerostipes, and Oscillospira), Actinobacteria (Bifidobacterium genus), Proteobacteria, Bacteroidetes (genera Bacteroides, Prevotella, and Fusobacterium) and a significant decrease in the phyla Bacteroidetes (family Muribaculaceae, and genera Bacteroides, Prevotella, Alistipes, Paludibacter, Barnesiella), Firmicutes (genera Faecalibacterium, Eubacterium, Lachnospira, and Roseburia), Actinobacteria (Bifidobacterium genus), Proteobacteria (Desulfovibrio genus). The influence of GSD on microbial diversity is not clear. Some studies report that GSD reduces microbial diversity in the bile; whereas others suggest the increase in microbial diversity in the bile of patients with GSD. The phyla Proteobacteria (especially family Enterobacteriaceae) and Firmicutes (Enterococcus genus) are most commonly detected in the bile of patients with GSD. On the other hand, the composition of bile microbiota in patients with GSD shows considerable inter-individual variability. The impact of GSD on the Firmicutes/Bacteroidetes ratio is unclear and reports are contradictory. For this reason, it should be stated that the results of reviewed studies do not allow for drawing unequivocal conclusions regarding the relationship between GSD and the Firmicutes/Bacteroidetes ratio in the microbiota.

  1. Page 1, Introduction. The author should write a few sentences on the role of gastrointestinal hormones, including ghrelin, in the regulation of food intake and obesity development, as well as protective and healing effects of ghrelin in the gut. For this purpose, the author could use the following interesting articles: Ceranowicz et al. (PMID: 25716961), Cui et al. (PMID: 28232667), Warzecha et al. (PMID: 22300084), Dembinski et al. (PMID: 14726611), Bukowczan et al. (PMID: 25594510), Matuszyk et al. (PMID: 27598133).

Response. You give a list of very interesting studies on ghrelin, but I cannot refer to them, because either the full text is not available [Warzecha et al. (PMID: 22300084), Bukowczan et al. (PMID: 25594510)] or the topics is far from this article  - investigate the influence of ghrelin on the development of acute pancreatitis in rat - Dembinski et al. (PMID: 14726611) and ischemia-reperfusion-induced pancreatitis in animals - Bukowczan et al. (PMID: 25594510), or acetic acid-induced colitis in rats - Matuszyk et al. (PMID: 27598133), and I refer only to two works - Ceranowicz et al. (PMID: 25716961), Cui et al. (PMID: 28232667). Sorry, I didn't put it in bold.

I added 33 lines (55-88). A significant relationship exists among food intake, energy balance and gut peptides that are secreted from gastrointestinal enteroendocrine cells, such as ghrelin, leptin, glucagon-like peptide-1, cholecystokinin (CCK), peptide tyrosine tyrosine (PYY), and serotonin [20]. Let's focus on two of them. Ghrelin, an orexigenic peptidyl hormone secreted from the stomach, was discovered in 1999 and is associated with feeding and energy balance [21]. Ghrelin increases appetite and energy expenditure and promotes the use of carbohydrates as a source of fuel at the same time as sparing fat [22]. The development of resistance to leptin and ghrelin, hormones that are crucial for the neuroendocrine control of energy homeostasis, is a hallmark of obesity [23]. The impact of acyl-ghrelin on glucose metabolism and lipid homeostasis may allow for novel preventative or early intervention therapeutic strategies to treat obesity-related type 2 diabetes and associated metabolic dysfunction [24]. There were no differences for total bile acids, insulin, ghrelin, and glucose-dependent insulinotropic polypeptide between patients with GSD and the control group without gallstones [25]. Mendez-Sanchez et al (2006) found an inverse correlation of serum ghrelin levels and the prevalence of GSD in a logistic regression analysis (OR = 0.27, 95%CI 0.09-0.82, P = 0.02) [26]. Authors suggest that serum ghrelin concentrations are associated with a protective effect of GSD and this is related to a motilin-like effect of ghrelin on the gallbladder motility. But the median of serum ghrelin values did not show a difference between the patients and controls (660 vs. 682 ng/L) [26]. 

Leptin is associated with obesity: although it should reduce food intake and body weight, in obese patients the serum leptin levels are higher than in the lean individuals and do not manage reducing their food intake [27].  insulin and leptin play an important role in the development of prediabetes and NAFLD, which is a risk factor for GSD. There could be the following pathogenic links: obesity promotes insulin resistance; high levels of insulin increase leptin levels; leptin cannot lead to decreased insulin levels and decreased appetite because of leptin resistance in the nervous system and in the adipose tissue; and high levels of leptin promote hepatic steatosis which in turn increases insulin resistance [27].  Positive correlations between serum leptin and BMI, CCK, total cholesterol, and insulin were found in the gallstone group [28].

Gut microbiota can regulate levels of these gut peptides and thus regulate intestinal metabolism via the microbiota-gut-brain axis [20]. Serum ghrelin levels were negatively correlated with Bifidobacterium, Lactobacillus and B. coccoides–Eubacterium rectale, and positively correlated with Bacteroides and Prevotella [29]. Leptin was negatively correlated with Clostridium, Bacteroides and Prevotella, and positively correlated with Bifidobacterium and Lactobacillus [29].  The results of the studies on the relationship between GSD, obesity, and incretin hormones remain controversial.

  1. The author should again carefully read the manuscript and try to correct still existing errors. For example, page 1 lines 34-35. The abbreviation “etc” can be used when presenting items or concepts belonging to one group with common distinct features. “Etc” means “and so on”. What are the common features of atherosclerosis, diabetes, and gallstone disease, and what other diseases belongs to the same group? The author should add “or” after diabetes and remove “etc”.

Response. I don't agree with you. all these diseases have fundamental common features! This is an axiom. I'm a gastroenterologist.

Many studies (more 6000 results in PubMed.gov) have reported that metabolic diseases, such as obesity, type 2 diabetes, dyslipidemia, atherosclerosis, coronary heart disease, hypertension, NAFLD, and gallstone disease are associated with disorders of lipid and carbohydrate metabolism: dyslipidaemia and insulin resistance. 

Bugajska J et al (2019) conclude that type 2 diabetes mellitus, coronary disease, and GSD might have common genetic and environmental antecedents [25].

Wang S-Z (2020) [20] (quote) “…gut microbiota can affect carbohydrate, lipid, and amino acid metabolism, and thus influence several metabolic diseases, such as obesity, type 2 diabetes, dyslipidemia, non-alcoholic fatty liver disease (NAFLD), gout, vitamin deficiency, and atherosclerosis [1,8,9,10]».

 “Interestingly, bile acid receptor activation of FXR and TGR5 can relieve metabolic diseases, such as obesity, type 2 diabetes, dyslipidemia, NAFLD, and atherosclerosis [163,164]”.

“…lead to liver damage and various inflammatory and metabolic diseases, such as alcoholic liver disease, NAFLD, primary biliary cholangitis, primary sclerosing cholangitis, and cirrhosis [173]”.

The keyword search “metabolic diseases and obesity and type 2 diabetes and atherosclerosis» - 1599 results, “metabolic diseases and obesity and diabetes and gallstones” – 121 results, “metabolic syndrome and gallstones” – 148 results in PubMed.gov.

Keywords: “atherosclerosis and gallstones” - 92 articles, “diabetes and gallstones”- 782 articles in PubMed.gov.

  1. Di Ciaula A, … Wang DQ, Lammert F, Portincasa P. The Role of Diet in the Pathogenesis of Cholesterol Gallstones. Curr Med Chem. 2019;26(19):3620-3638. PMID: 28554328. Line 3-4 of the abstract (quote): “Cholesterol gallstone disease is a major health problem in Westernized countries and depends on a complex interplay between genetic factors, lifestyle and diet, acting on specific pathogenic mechanisms. Overweigh, obesity, dyslipidemia, insulin resistance and altered cholesterol homeostasis have been linked to increased gallstone occurrence”.
  2. Méndez-Sánchez N, et al. Gallstones are associated with carotid atherosclerosis.Liver Int. 2008 Mar;28(3):402-6. PMID: 18069975
  3. Di Ciaula A, Wang DQ, Portincasa P. An update on the pathogenesis of cholesterol gallstone disease. Curr Opin Gastroenterol. 2018 Mar;34(2):71-80. PMID: 29283909. Line 7-8 of the abstract (quote): “Mechanisms of disease are linked with insulin resistance, obesity, the metabolic syndrome, and type 2 diabetes”. 
  4. Smelt AH. Triglycerides and gallstone formation. Clin Chim Acta. 2010 Nov 11;411(21-22):1625-31. PMID: 20699090. the abstract (quote): “Patients with hypertriglyceridemia (HTG) - often overweight and insulin resistant - are at risk for gallstone disease. Peroxisome proliferator-activated receptors (PPARs), liver X receptors (LXRs), farnesoid X receptor (FXR) and hepatocyte nuclear factor 4α (HNF4α) are the nuclear receptors involved in the regulation of lipogenesis. Microsomal triglyceride transfer protein (MTP) is involved in the production of VLDL and its activation is also under control of transcription factors as FXR and Forkhead box-O1 (FoxO1). Triglyceride and BA (bile acid) metabolism are linked. There is an inverse relationship between bile acid fluxes and pool size and VLDL production and SHP (small heterodimer partner) and FXR are the link between BAs and TG metabolism. BAs are also ligands for FXR and G-protein-coupled receptors, such as TGR5. FXR activation by BAs suppresses the expression of MTP, transcription factor sterol regulatory element binding protein (SREBP)-1c and other lipogenic genes. LXRs stimulate lipogenesis whereas FXRs inhibit the metabolic process. Gallbladder motility is impaired in HTG patients compared to BMI matched controls. There is evidence that the gallbladder in HTG is less sensitive to CCK and that this sensitivity improves after reversal of high serum TG levels by use of TG lowering agents”.
  5. Chen Y, et al. FMO3 and its metabolite TMAO contribute to the formation of gallstones. Biochim Biophys Acta Mol Basis Dis. 2019 Oct 1;1865(10):2576-2585. doi: 10.1016/j.bbadis.2019.06.016. the abstract (quote): “TMAO may contribute to the development of diseases such as atherosclerosis because of its role in regulating lipid metabolism. In this study, we found that high plasma TMAO levels were positively associated with the presence of gallstone disease in humans”. 
  6. Di Ciaula A, Portincasa P. Recent advances in understanding and managing cholesterol gallstones. F1000Res. 2018 Sep 24;7:F1000 Faculty Rev-1529. PMID: 30345010. Line 4-5 of the abstract (quote): “From an epidemiologic point of view, the risk of gallstones has been associated with higher risk of incident ischemic heart disease, total mortality, and disease-specific mortality (including cancer) independently from the presence of traditional risk factors.”
  7. Wirth J, et al. Presence of gallstones and the risk of cardiovascular diseases: The EPIC-Germany cohort study. Eur J Prev Cardiol. 2015 Mar;22(3):326-34. doi: 10.1177/2047487313512218. ABSTRACT BACKGROUND: Gallstones are common disorders associated with several cardiovascular risk factors. Gallstone formation and atherosclerosis may share key pathways.
  8. Portincasa P, et al. Management of gallstones and its related complications. Expert Rev Gastroenterol Hepatol. 2016;10(1):93-112. doi: 10.1586/17474124.2016.1109445. Epub 2015 Nov 11. PMID: 26560258. Line 3-5 of the abstract (quote): “For cholesterol gallstone disease, moreover, a strong link exists between this disease and highly prevalent metabolic disorders such as obesity, dyslipidemia, type 2 diabetes, hyperinsulinemia, hypertriglyceridemia and the metabolic syndrome.
  9. Di Ciaula A, Wang DQ, Portincasa P. Cholesterol cholelithiasis: part of a systemic metabolic disease, prone to primary prevention.Expert Rev Gastroenterol Hepatol. 2019 Feb;13(2):157-171. doi: 10.1080/17474124.2019.1549988. the abstract (quote): “Cholesterol gallstone disease have relationships with various conditions linked with insulin resistance, but also with heart disease, atherosclerosis, and cancer.”
  10. Sang JH, et al. Correlations between metabolic syndrome, serologic factors, and gallstones. J Phys Ther Sci. 2016 Aug;28(8):2337-41. doi: 10.1589/jpts.28.2337.
  11. Zhu Q, et al. The association between gallstones and metabolic syndrome in urban Han Chinese: a longitudinal cohort study.Sci Rep. 2016 Jul 22;6:29937. doi: 10.1038/srep29937.
  1. The author wrote: “Risk factors of the GSD is age...” What kind of age? Young age, average age or old age? Moreover, in the case of risk factors “is” should be replaced by “are”.

Response. I've been researching to risk factors for gallstone disease for over 20 years. This is a generally accepted designation in the world literature of age as a risk factor for the development of GSD. I will give some examples, in particular, of our luminaries (Lammert, Portincasa, van Erpecum, Wang et al) and recent studies. Although this means, of course, "increasing the age", as well as “BMI” - increasing the BMI, “pregnancy” - increasing the number of pregnancies.

  • Lammert F, Gurusamy K, Ko CW, Miquel JF, Méndez-Sánchez N, Portincasa P, van Erpecum KJ, van Laarhoven CJ, Wang DQ. Nat Rev Dis Primers. 2016 Apr 28;2:16024. doi: 10.1038/nrdp.2016.24. Line 8-9 of the abstract (quote): “Risk factors for gallstones are female sex, age, pregnancy, physical inactivity, obesity and overnutrition.”
  • Chen CY, et al. Age is one of the risk factors in developing gallstone disease in Taiwan. Age Ageing. 1998 Jul;27(4):437-41. doi: 10.1093/ageing/27.4.437. Line 17-18 of the abstract (quote): “ConclusionsAge, high body mass index, diabetes mellitus and glucose intolerance are the risk factors for developing GSD in Taiwan.”
  • Panpimanmas S, Manmee C. Risk factors for gallstone disease in a Thai population. J Epidemiol. 2009;19(3):116-21. doi: 10.2188/jea.je20080019. Epub 2009 Apr 28.Line 6-7 of the abstract (quote): “The risk factors examined were age, sex, BMI, use of oral contraceptives, diabetes mellitus, …”. 
  • Chen YC, Chiou C, Lin MN, Lin CL. The prevalence and risk factors for gallstone disease in taiwanese vegetarians. PLoS One. 2014 Dec 18;9(12):e115145. doi: 10.1371/journal.pone.0115145. Line 17-18 of the abstract (quote): “ConclusionsRisk factors useful for predicting GSD in vegetarians are (1) age and total bilirubin level in men, and (2) age, BMI, and alcohol consumption in women”.
  • Martínez García RM, et al. Intervención nutricional en el control de la colelitiasis y la litiasis renal [Nutritional intervention in the control of gallstones and renal lithiasis]. Nutr Hosp. 2019 Aug 27;36(Spec No3):70-74. Spanish. doi: 10.20960/nh.02813. Line 2-4 of the abstract (quote): “Age, female sex, … are factors associated with an increased risk of cholelithiasis”. 
  • Aldriweesh MA, et al. The Incidence and Risk Factors of Cholelithiasis Development After Bariatric Surgery in Saudi Arabia: A Two-Center Retrospective Cohort Study. Front Surg. 2020 Oct 22;7:559064. doi: 10.3389/fsurg.2020.559064. Line 14-15 of the abstract (quote): “Gender, age, and comorbidities were not associated with the formation of cholelithiasis”.

In the case of risk factors (line 40), I replaced “ is "with "are", thank you.

  1. Page 3, line 83. The author wrote: “Other important functions of Bacteroides spp. include the growth of mucus”. Did the author want to write about mucus or mucosa? What does growth of mucus mean? Did the author want to write “the stimulation of mucus production or release” or stimulation of mucosa growth’?

Response. I agree with You, I deleted this fragment.

  1. Page 4, lines163-166. The sentence is too long and unclear. This must be corrected, and the some comments about “ect” abbreviation as in the comment 3.

Response. I agree with You, I corrected this sentence.

Lines 200-204. “It is proved that intestinal dysbiosis makes a significant contribution to the development of not only the GSD itself [5, 6, 71-74], but also to the development of numerous disorders that are risk factors for GSD, including obesity [21-32], type 2 diabetes [75], hypercholesterolemia [40, 43], diet [76], NAFLD [77-80], cardiovascular diseases [60, 81], physical inactivity [82-84], etc.”

I would like to express my gratitude to the editorial board of the journal "Journal of Personalized Medicine" for their attention to our article.

Sincerely Yours, Irina Grigor’eva.